# Dietary Interventions for Gout and Effect on Cardiovascular Risk Factors: A Systematic Review

**DOI:** 10.3390/nu11122955

**Published:** 2019-12-04

**Authors:** Daisy Vedder, Wendy Walrabenstein, Maaike Heslinga, Ralph de Vries, Michael Nurmohamed, Dirkjan van Schaardenburg, Martijn Gerritsen

**Affiliations:** 1Amsterdam Rheumatology and Immunology Center|Reade, 1056 AB Amsterdam, The Netherlands; m.heslinga@reade.nl (M.H.); m.nurmohamed@reade.nl (M.N.); d.v.schaardenburg@reade.nl (D.v.S.); m.gerritsen@reade.nl (M.G.); 2Amsterdam UMC|Amsterdam Medical Center, 1105 AZ Amsterdam, The Netherlands; 3Medical Library, Vrije Universiteit, 1081 HV Amsterdam, The Netherlands; r2.de.vries@vu.nl; 4Amsterdam UMC|VU Medical Center, 1081 HV Amsterdam, The Netherlands

**Keywords:** gout, hyperuricemia, cardiovascular disease, metabolic syndrome x, diet, purine low diet, Mediterranean diet, DASH, cholesterol, blood pressure

## Abstract

Gout is one of the most prevalent inflammatory rheumatic disease. It is preceded by hyperuricemia and associated with an increased risk for cardiovascular disease, both related to unhealthy diets. The objective of this systematic review is to better define the most appropriate diet addressing both disease activity and traditional cardiovascular risk factors in hyperuricemic patients. We included clinical trials with patients diagnosed with hyperuricemia or gout, investigating the effect of dietary interventions on serum uric acid (SUA) levels, gout flares and—if available—cardiovascular risk factors. Eighteen articles were included, which were too heterogeneous to perform a meta-analysis. Overall, the risk of bias of the studies was moderate to high. We distinguished four groups of dietary interventions: Calorie restriction and fasting, purine-low diets, Mediterranean-style diets, and supplements. Overall, fasting resulted in an increase of SUA, whilst small (SUA change +0.3 to −2.9 mg/dL) but significant effects were found after low-calorie, purine-low, and Mediterranean-style diets. Studies investigating the effect on cardiovascular risk factors were limited and inconclusive. Since Mediterranean-style diets/DASH (Dietary Approach to Stop Hypertension) have shown to be effective for the reduction of cardiovascular risk factors in other at-risk populations, we recommend further investigation of such diets for the treatment of gout.

## 1. Introduction

Gout is one of the most common inflammatory joint disease and its prevalence is estimated at between 0.1% and 10% and is rising, especially in Western countries [1]. For the development of gout, hyperuricemia causing monosodium urate deposition in tissues is mandatory. When gout is not treated adequately it evolves into a chronic and disabling disease. Gout is preceded by hyperuricemia, which is defined by a serum urate (SUA) above >7 mg/dL. Hyperuricemia results most often from reduced renal excretion of urate, frequently combined with a high intake of purines [2]. There is a strong association between SUA level and the risk of developing gout: Subjects with hyperuricemia have a more than 30 times higher risk of developing gout than persons with normal levels [3].

Risk factors for developing hyperuricemia and gout include non-modifiable risk factors (sex, age, race, and genetics) and modifiable risk factors, such as diet and lifestyle [4]. Cohort studies have shown that a typical Western diet (based on a high intake of red and processed meats, refined grains, and sugar-rich meals) is associated with a 42% higher risk of developing gout, whilst adherence to a Mediterranean-type diet is associated with a lower risk (RR 0.68, 95% confidence interval (CI) 0.57 to 0.80, *p* < 0.001) [5]. Furthermore, patients with obesity (BMI > 25 kg/m^2^) have a two- to three-fold increased risk of gout compared to patients with a BMI below 25 [6].

Gout is associated with an increased risk of cardiovascular disease. This is mainly due to a high prevalence of traditional cardiovascular risk factors [7,8]. A recent cross-sectional study among 237 gout patients showed that 92% of these patients had at least one traditional cardiovascular risk factor (mostly hypertension and dyslipidemia) [7].

Besides this high prevalence of traditional risk factors, gout is also an independent risk factor for cardiovascular morbidity and mortality independent of other measured risk factors, with hazard ratios for mortality due to coronary heart disease (CHD) and cardiovascular disease (CVD) of 1.4 (95% CI 1.2–1.6) and 1.3 (95% CI 1.2–1.4), respectively, after adjusting for traditional cardiovascular risk factors [9]. Gout-specific characteristics associated with an increase in cardiovascular risk are the presence of high SUA levels (>9.1 mg/dL), a longer disease duration (≥2 years), polyarticular disease, joint damage, and tophi [10].

The European League Against Rheumatism (EULAR) recommendations for the treatment of gout recommend lifestyle advice for all gout patients. Advice on diet includes the avoidance of alcohol, sugar-sweetened drinks, heavy meals, and excessive intake of meat and seafood, reflecting a purine-low diet [11]. This diet seems an obvious choice for the management of hyperuricemia and gout, since urate is generated through the degradation of purines. Increased intake of purine-rich products as meat is indeed associated with higher SUA levels. To a lesser extent, the same applies to seafood but not to legumes, both also rich in purines. Legumes are high in fiber, which could bind to uric acid in the intestines before excretion [12].

Given the many hypotheses and the limited evidence, we aimed to get a clear picture of the current state of knowledge and the gaps in this systematic review on the effect of dietary interventions on SUA levels and any other outcome related to metabolic syndrome and cardiovascular disease in patients with hyperuricemia or gout. The ultimate objective is to define an appropriate diet for hyperuricemia and gout, one that addresses both gout activity as well as the traditional cardiovascular risk factors.

## 2. Materials and Methods

This systematic review is based on the Preferred Reporting Items for Systematic Reviews and Meta-Analyses (PRISMA) method [13]. Two authors, D.V. and R.d.V., searched Wiley/Cochrane Library, PubMed, Ebsco/CINAHL, and Embase.com up to 1 March 2019. The following key words were used (including synonyms and closely related words) as index terms or free-text words: “Gout”, “hyperuricemia”, “nutrition therapy”, “diet”, “uric acid”, and “cardiovascular diseases”. The search was performed without date, language, or publication status restriction. All duplicate articles were excluded. The full search strategies for all databases can be found in the Appendix A. We included (randomized) clinical trials with patients diagnosed with asymptomatic hyperuricemia (SUA >7 mg/dL in men and >6 mg/dL in women) or gout and aged 18 years or older. We included studies investigating the effect of dietary interventions (including supplements) on SUA levels, gout incidence, gout flares and—if available—cardiovascular risk factors/events when compared with other diets, supplements, or standard care. Regarding cardiovascular risk factors, we verified all studies on the effects of the intervention on weight (kg) or body mass index (kg/m^2^), blood pressure (mmHg), lipid profile (total cholesterol, high-density lipoprotein (HDL), low-density lipoprotein (LDL), triglycerides in mmol/L), blood glucose (mmol/L), metabolic syndrome prevalence, and incidence or prevalence of cardiovascular events and cardiovascular disease. Cohort studies, case reports, animal studies, and articles written in a language other than English were excluded from this review.

Two authors, D.V. and M.H., independently assessed and selected studies eligible for this systematic review based on the title and abstract. Disagreement was resolved by consensus. Two authors, D.V. and W.W., read the full texts of the remaining articles and made a final selection. Data were extracted and presented in a table, including author, title, year, journal, study design, date of data abstraction, description population, age, inclusion and exclusion criteria, sample size, intervention method and duration of follow-up, use of control group, primary and secondary outcomes, control for confounding and effect modification, conclusion, and limitations. The included articles were assessed for the risk of bias using the Cochrane Collaboration’s tool (RCT’s) and the ROBINS-I (the Risk Of Bias In Non-randomized Studies-of Interventions) assessment tool. Due to the large variation in the duration and intervention and control treatments, it was decided not to conduct a meta-analysis.

## 3. Results

The search resulted in 2784 potential articles from PubMed (*n* = 939), Wiley/Cochrane library (*n* = 119), CINAHL (*n* = 138), and Embase (*n* = 1588). Two additional articles were found through references. In total, 2741 articles were excluded based on double-blind screening of the title and abstract, including 680 duplicates. The remaining 45 full text articles were assessed, and 18 articles were found eligible for this systematic review. See Figure 1 for the flowchart of the search and selection process. Overall, the risk of bias of both randomized and non-randomized studies was moderate to high, as shown in Figure 2 and Figure 3.

All included articles are summarized in Table 1. The following two paragraphs are dedicated to the effects of the studied interventions on (1) SUA levels and (2) cardiovascular risk factors (blood pressure, lipid profile, glucose, and weight). No studies were found regarding the incidence of cardiovascular events/disease. Most studies had SUA as the primary outcome. The various dietary interventions were categorized in four groups: (a) Calorie restriction and fasting, (b) purine-low diets, (c) Mediterranean-style diets, and (d) supplements.

### 3.1. Effect of Diet Interventions on Serum Uric Acid Level and Gout Flares

#### 3.1.1. Calorie Restriction and Fasting

Six studies were found investigating the effect of diet interventions with different grades of calorie restriction or fasting. Maclachlan et al. (1967) studied the effects of combinations of a purine-low diet, fasting, and alcohol intake in a 6-week study [14]. Nine gouty subjects (*n* = 4 alcoholics) were admitted to hospital. SUA at baseline was measured after urate-lowering therapy (ULT) was discontinued and ranged from 7.2 to 13.9 mg/dL. This level decreased after a 2-week purine-low diet (average = −1.4 mg/dL). After one day of fasting, all subjects showed a rise in SUA levels, with a mean difference of 1.1 mg/dL (1-day fasting) and 2.0 mg/dL (2-day fasting), which returned rapidly to near baseline values after 24 h of refeeding. During a second period of fasting, subjects used alcohol (79–112 g, whiskey) on the two consecutive days of fasting. The mean average increase in SUA in the subjects with gout was 2.4 mg/dL compared to 2.0 mg/dL on fasting without alcohol. Increases in SUA were accompanied by decreases in urate excretion. The subjects experienced 24 gout attacks in total, with most of these attacks (*n* = 19) directly following an increase of SUA of at least 1 mg/dL.

Scott et al. (1977) focused on the effect of a low-calorie and low-carbohydrate diet, which was not further defined, in 20 obese subjects (>10% above ideal body weight) until ‘adequate’ weight reduction was achieved (duration unknown) [15]. Not all subjects were diagnosed with gout or hyperuricemia (no numbers given). A mean weight loss of 7.0 kg was accompanied by a mean decrease in plasma urate of 0.5 mg/dL (*p* = 0.007) and lower levels of urinary urate.

Yamashita et al. (1986) performed a pilot study in 27 overweight subjects (BMI >26 kg/m^2^) of whom 19 (70%) met the criteria for hyperuricemia (gout prevalence unknown) [16]. The pilot study included hospitalization of the subjects and a gradual weekly reduction of total calorie intake (1500–1200–1000–800 kcal/day) with a parallel reduction in dietary purine intake from 154 to 116 mg/day. The 800-kcal diet was kept constant afterwards until the end of the study (mean duration 9 weeks). Food intake was checked by doctors and nurses. Exercise therapy was added to the regime after the reduction of total calorie intake to 800 kcal/day. The mean weight loss at time of discharge (up to 14 weeks) was 7% to 8% of the initial body weight. This was accompanied by a decrease in urate of 1.3 mg/dL in females and 1.8 mg/dL in male subjects (*p* < 0.01). The authors concluded that a gradual reduction in kilocalories led to a decrease in SUA levels.

Tinahones et al. (1995–1997) examined the effect of a three-week 1200-kcal diet followed by a 3-week 2500-kcal diet in 36 subjects with hyperuricemia [17,18]. Both diets contained 50% of energy (en%) carbohydrates, 20 en% protein, and 30 en% fat. Subjects were advised to avoid foods with a purine level >75 mg/100 g. The group was divided into 20 subjects with hyperuricemia (group I) and 16 subjects with both hyperuricemia and hypertriglyceridemia (group II). The mean weight loss after the intervention in group I was 4.0 kg and 4.8 kg in group II. After three weeks on the 1200-kcal diet, the SUA levels dropped non-significantly with 0.63 and 1.28 mg/dL in groups 1 and 2, respectively. SUA increased again to levels still below baseline after three weeks on the 2500-kcal diet.

The study of Dessein et al. (2000) only included subjects with gout [19]. Thirteen subjects with recurrent gout (>two self-reported attacks in the last 4 months) and obesity started with a 1600-kcal diet (40 en% carbohydrates, 30 en% protein) for 16 weeks. Weight loss and a decrease in the frequency of gout attacks occurred in all except one subject. Mean weight loss compared to baseline was 8% (*p* = 0.002) and the frequency of monthly attacks decreased from 2.1 to 0.6 (*p* = 0.001). The mean SUA decrease after the intervention was 1.69 mg/dL and SUA normalized in 7 of the 12 subjects with hyperuricemia (58%). Further follow-up of nine subjects showed no significant change in BMI and SUA after 12 months compared to week 16 of the intervention.

The last study on fasting was performed by Habib et al. (2014) [20]. They examined the effect of Ramadan (fasting, including no drinking from sunrise to sunset, for one month) on SUA and gout activity in patients from an Israeli hospital with gout and compared results with those in subjects with gout who did not participate in the fasting period of Ramadan. Forty-three subjects completed the study and no significant difference was found in adherence to a low-purine diet, gouty activity, and SUA between both groups. Among the fasting subjects, there were no signs of dehydration, which is thought to be the main mechanism leading to a suspected SUA rise during Ramadan.

#### 3.1.2. Purine-Low Diet

Two intervention studies were found examining the effect of a purine-low diet on SUA and one study examining the effect of dietary education of subjects on the beneficial effects of a purine-low diet on SUA and gout activity. The study of Peixoto et al. (2001) investigated the effect of a purine-low diet in 55 subjects with hyperuricemia and hypertension [21]. The diet recommended avoidance of foods with purine >100 mg/100 g products and a decrease in the consumption of foods with moderate purine contents (9–100 mg purine/100 g food). Intake (kcal, macronutrients, purine, oxalic acid) was measured with a 24-h recall and a semi-quantitative food frequency questionnaire. The effect of diet (group 1) was compared with the effects of (group 2) subjects treated with a purine-low diet combined with allopurinol (150–300 mg) and (group 3) subjects receiving medication only without a diet intervention. SUA levels were significantly reduced at week 12 in all three groups (−1.2, −2.5, and −2.4 mg/dL, respectively, *p* < 0.001), without any statistical difference between groups. At week 24, a further decrease in SUA was only seen in group 1 (−2.1 mg/dL compared to baseline, *p* < 0.001). SUA in groups 2 and 3 slightly increased in the period between 12 and 24 weeks, although SUA at 24 weeks was still significantly lower compared to baseline (*p* < 0.05). No information was given on medication adherence in groups 2 and 3.

Cardona et al. (2005) focused on subjects with gout and currently not on ULT and implemented a low-purine diet for two weeks [22]. Subjects were instructed to avoid certain foods, such as meat, poultry, fish, alcohol, and some legumes/vegetables. Sixty-four subjects with gout participated, with a mean SUA level of 7.7 mg/dL at baseline. After two weeks following a purine-low diet, urate decreased to 0.57 mg/dL (*p* = 0.01).

Holland et al. (2015) studied the effect of dietary advice on 29 subjects with gout on a stable dose of ULT (allopurinol 100–900 mg/day) [23]. The control group received advice at baseline regarding the importance of compliance with drug therapy, the benefit of weight loss and exercise, and the benefit of reduced alcohol intake. On top of that, the intervention group received comprehensive dietary advice based on the guidelines provided by the British Society of Rheumatology (restriction on high-purine foods, restricted intake of protein, and an emphasis on the importance of weight loss, not further defined) [33] as well as specific guidelines on alcohol, use of sugar, and higher intakes of healthy foods, such as dairy products, vegetables, nuts, legumes, fruits, and whole grains, based on the recommendations by Choi (2010) [34]. At baseline, the mean SUA was 4.88 mg/dL in both groups. At six months, there was a significant difference in diet modification and knowledge of gout (based on the number of correct responses on a 13-item questionnaire regarding cause, gout attacks, medication, target SUA level, and diet) in favor of the intervention group (respectively *p* = 0.009, *p* < 0.001) but no significant difference in SUA (intervention group 5.04 mg/dL versus control group 4.54 mg/dL, *p* > 0.05).

#### 3.1.3. Mediterranean-Style Diets

In the last five years, more studies were published on the efficacy of more plant-based diets with less meat, fish, and dairy products. The Mediterranean diet and the Dietary Approach to Stop Hypertension (DASH) are examples favoring the use of fruits, vegetables, legumes, olive oil, nuts, and whole grains. Consumption of wine, dairy products, and poultry is moderate and low consumption of red meat, sweet beverages, creams, and pastries is advised [35].

Chatzipavlou et al. (2013) performed a pilot study in 12 subjects with hyperuricemia (male SUA >7.0 mg/dL, female >5.7 mg/dL) investigating the effect of a Cretan-style Mediterranean diet (1800–2400 kcal, rich in monounsaturated fatty acids (MUFAs, e.g., olive oil), legumes, cereals/bread, fruit, and vegetables; moderate in alcohol and dairy; and low in meat) on SUA [24]. Six subjects (50%) followed the diet for at least 8 weeks. Mean SUA at baseline was 9.12 (range 8.2–13.8) mg/dL and decreased to 6.2 (range 5.5–7.8) mg/dL in the six subjects. No gout attacks were experienced during the intervention.

Zhang et al. (2016) compared a standard diet according to dietary guidelines, with a diet enriched with soybeans and fruit products in a 3-month randomized controlled trial with 187 subjects with hyperuricemia [25]. Subjects with hypertension, diabetes mellitus, and subjects using ULT were excluded. After three months, no significant difference was found in daily food intake between groups. Surprisingly, and not intended, the consumption of fruits and soybean products increased equally in both the intervention and control group. After three months, both groups showed a significant decrease in serum urate, −1.0 and −1.1 mg/dL, respectively (*p* < 0.001 within both groups).

Tang et al. (2017) performed a randomized controlled trial on the effect of the DASH in subjects with pre- or stage 1 hypertension [26]. They included 103 subjects in their RCT, 24 of them with hyperuricemia (SUA ≥6.0 mg/dL). After a two-week run-in period on an isocaloric ‘American’ (control) diet, they were randomized to either continue the control diet or start the DASH. While SUA remained stable in the control group (−0.1 mg/dL), a decrease of −1.0 mg/dL was seen in the intervention group after 90 days (*p* = 0.03). The same research group (Juraschek et al., 2018) also conducted an 8-week study among African Americans with controlled hypertension, categorized in five baseline SUA groups (≤5, >5–6, >6–7, >7–8, and >8 mg/dL) [27]. Of 117 subjects, 64 had a baseline SUA >6 mg/dL. In this superiority study, DASH (including 240 USD in a debit account to purchase foods) with and without additional coaching was compared. No difference was found between or in the two groups after 8 weeks, although a significant trend was found toward greater SUA reduction among subjects with higher baseline SUA, with a decrease of −1.02 (−2.37, 0.32) in 18 subjects with a baseline SUA >8 mg/dL (*p* = 0.008).

#### 3.1.4. Supplements

Five studies examined the effect of nutrition supplements on serum urate in subjects with hyperuricemia or gout. Dalbeth et al. (2012) examined the effect of skim milk powder on the frequency of gout flares in 120 patients who met the American College of Rheumatology (ACR) criteria for gout [28]. In this randomized double-blind controlled trial, patients received (1) skim milk, (2) skim milk enriched with glycomacropeptide (GMP) and G600 milk fat extract, or (3) skim milk enriched with lactose powder. The frequency of gout flares decreased significantly in all three groups, without any difference between the groups. SUA did not change significantly.

Stamp et al. (2013) performed an open label, parallel group, randomized controlled trial on patients with gout and SUA >6 mg/dL [29]. Of the 40 included patients, 20 were not on ULT (groups 1 and 2) and 20 used ULT (groups 3 and 4). The four groups received the following treatment: (1) Daily supplement of 500 mg vitamin C, (2) allopurinol (50–100 mg daily), (3) increase current dose of allopurinol with 50 to 100 mg daily, and (4) add 500 mg of vitamin C to the current dose of allopurinol (REF). At the end of the 8-week duration of this study, the allopurinol groups 2 and 3 showed significant superiority, with lower SUA compared to the vitamin C groups 1 and 4 (*p* < 0.05).

A second study on the effect of vitamin C was performed by Azzeh et al. (2017). They investigated the effect of a daily 500-mg vitamin C supplement on SUA in subjects with hyperuricemia or gout [31]. In this study, 30 subjects were included, 15 with gout and 15 with hyperuricemia. After 8 weeks, the SUA level insignificantly increased (+0.31 mg/dL, *p* > 0.05) in subjects with gout and significantly decreased in the group with asymptomatic hyperuricemia (−0.78 mg/dL, *p* < 0.05).

Kubomura et al. (2016) conducted a double-blind placebo-controlled study investigating the effect of tuna extract on SUA in subjects with asymptomatic hyperuricemia [30]. The rationale behind this supplementation was that imidazole compounds, as found in tuna extract, might affect the acidity of body fluids, renal functions, and organic acid levels associated with reabsorption of urate in the glycolytic pathway. The subjects were divided in three groups: (1) Placebo, (2) low-dose tuna extract (239 mg/day), and (3) high-dose tuna extract (477 mg/day). Urate levels in all groups decreased compared to baseline in week 4 (0.07, 0.23, and 0.34 mg/dL, respectively). An additional measurement two weeks after the intervention showed that SUA in the high-dose tuna extract group had further decreased, with −0.49 mg/dL compared to +0.14 mg/dL in the placebo group from baseline (*p* < 0.05).

Yamanaka et al. (2018) recently performed a randomized, double-blind, placebo-controlled trial, which investigated the ability of *Lactobacillus delbrueckii* ssp. bulgaricus and *Streptococcus thermophilus* (hereafter summarized as PA-3) to decrease SUA in 25 subjects with hyperuricemia or gout [32]. ULT was discontinued 4 weeks before the intervention. Subjects subsequently received yoghurt beverages with PA-3 or without PA-3 (placebo) twice a day (2 × 100 g) for 8 weeks. SUA did not change in the intervention group (0 mg/dL) and increased in the control group (+ 0.2 mg/dL, not significant).

### 3.2. Diet Intervention and Changes in Cardiovascular Risk Factors

The studies in which cardiovascular risk factors were also considered are listed in Table 2 with a summary of the outcomes for blood pressure, lipid profile, glucose, and body weight. All diet categories contained studies covering cardiovascular risk factors, with the exception of supplements.

#### 3.2.1. Calorie Restriction and Fasting

As expected, calorie-restricted interventions resulted in significant weight loss. Although the duration and amount of calorie restriction (800–1600 kcal) between studies differed, the decrease in weight was approximately 5% to 8% of baseline body weight (Dessein, Tinahones, Scott, Yamashita). Scott, Tinahones, and Dessein also showed significant decreases in total cholesterol and triglycerides, which was accompanied by slight changes in HDL. Dessein et al. also reported a significantly lower LDL (baseline 3.5 ± 1.2; end 2.7 ± 0.8 mmol/L, *p* = 0.004) after 16 weeks. All lipid levels in the study of Tinahones returned to values close to baseline levels upon completion of the 2500-kcal diet. No data were published on blood pressure and glucose in the studies on calorie restriction and fasting.

#### 3.2.2. Purine-Low Diets

Cardona (2005) and Peixoto (2001) et al. investigated the effect of a purine low diet on weight, total cholesterol, triglycerides, and glucose. Weight remained stable in both studies. Total cholesterol and triglycerides increased slightly after 12 weeks in the study by Peixoto (not significant) and decreased in the 2-week study by Cardona (*p* < 0.001 for total cholesterol and triglycerides). Glucose at baseline was slightly increased in the study by Cardona et al. (6.0 ± 1.1 mmol/L) and improved significantly within 2 weeks (−0.3 ± 0.7, *p* < 0.001), whereas the subjects in the study by Peixoto et al. had lower baseline glucose levels, which also decreased though not significantly. Peixoto also looked into the effect of the 12-week purine-low diet on blood pressure. Only group 1 (purine-low diet without medication) showed significant decreases of both systolic as well as diastolic blood pressure (baseline 150 ± 22.6/102 ± 17.3 mmHg; end 133 ± 21.7/92 ± 12 mmHg, *p* < 0.05). It should be mentioned that all patients were known to have hypertension and on a stable dose of antihypertensive medication during the intervention. Physical activity and smoking did not change during the study.

#### 3.2.3. Mediterranean-Style Diets

In the study by Tang et al., no data on cardiovascular risk factors were reported for the hyperuricemic subgroup. The six subjects who completed the study of Chatzipavlou et al. on the effect of a Cretan Mediterranean diet showed an average 4% weight loss (BMI 31.46 (28.2–37.3) kg/m^2^ vs. 29.4 (26–34.2) kg/m^2^) after 8 weeks. This was accompanied by a slight decrease in total cholesterol (−0.3 mmol/L), triglycerides (−0.5 mmol/L), and LDL (−0.3 mmol/L). Statistical significance was not tested due to the small population size, and no data on blood pressure and glucose were published.

The Japanese subjects in the high fruit and soybean product diet (Zhang et al.) had a lower mean BMI at baseline (both groups BMI 25 kg/m^2^) than the Greek subjects in the study by Chatzipavlou and lost some weight, which was only significant (BMI: −0.5 kg/m^2^, *p* < 0.05) in group 1 (standard diet, not the fruit and soybean product group). In the intervention group, a small significant decrease was seen in total cholesterol and triglycerides, whilst no changes were seen in HDL in contrary to a small increase of HDL in the control group. The subjects were normotensive and not (pre-) diabetic at baseline and showed no change in blood pressure and glucose during the intervention.

## 4. Discussion

Diet interventions, including low-calorie diets (but not fasting), purine-low diets, and different variations of the Mediterranean diet, were able to decrease SUA in patients with asymptomatic hyperuricemia or gout. The range in SUA change after the interventions (excluding fasting) was +0.3 to −2.9 mg/dL. Most studies included in this review reported baseline SUA between 6.5 and 9.7 mg/dL According to the EULAR guideline, this implicates a required decrease of 0.5 to 3.7 mg/dL to achieve target levels (11). Only the DASH in the study by Tang et al. was able to reach clinically significant SUA levels below 6 mg/dL.

In addition to the small effect size, the quality of the studies overall was low to moderate due to insufficient attention given to confounding variables (e.g., exercise), small groups, absence of control groups, insufficient information on intervention methods, lack of distinction between intervention and control dietary intervention, no information on adherence to the intervention, and vagueness about dropouts.

A more detailed view on the content of the three diet categories (calorie restriction, purine low, Mediterranean) reveals that the diets look very different at first sight but also have characteristics in common, such as less animal protein, less saturated fatty acids, and less alcohol, resulting in a less energy-dense and thus lower-calorie diet. Although both the purine-low and the Mediterranean diets are low in animal-based purines, they differ in their levels of plant-based purine from legumes and vegetables. This could be an important difference, as observational studies have shown that in fact, higher intakes of legumes are associated with a significantly lower risk of gout [36,37]. In addition, Mediterranean-style diets contain more vegetables, fruit, and whole-wheat products, with an expected higher intake of fiber. High intake of dietary fibers (or acetate, a short chain fatty acid and metabolic product of fiber) had a positive effect on the resolution of a gout flare in an experimental model of mice with gout [38]. No effect was described on SUA.

An optimal diet for this patient group not only reduces SUA but also reduces cardiovascular risk factors. In this review, the number of studies investigating the effect of the different diet interventions on cardiovascular risk outcomes was little and again the quality was low to moderate. Furthermore, it should be noted that some interventions were quite demanding and difficult to implement in the daily life of gout patients not admitted to hospital.

Calorie restriction seems to be an effective strategy for weight loss (5% to 8% of initial body weight) and improvement of the lipid profile. The long-term feasibility of a low caloric intake is, however, questionable. Mediterranean-style diets also seem to induce weight loss (0%–4%) and improve lipid profiles. Blood pressure was measured in only two studies, with one study showing no effect (Zhang et al.). The Peixoto study on the effect of a purine-low diet versus medication showed a significant decrease in both systolic and diastolic blood pressure in the diet without medication group. Studies of purine-low diets showed either no further data, no effect, or conflicting results.

Most recent studies on the prevention of cardiovascular disease in other ‘at-risk’ populations (e.g., patients with diabetes or hypertension) focus on the DASH, which has showed significant reduction in both systolic (−11.4 mmHg, 95%CI −15.9 to −6.9, *p* < 0.001) and diastolic blood pressure (−5.5 mmHg, 95%CI −8.2 to −2.7, *p* < 0.001) among individuals with and without hypertension [39]. Furthermore, adherence to the DASH has shown a reduction of LDL cholesterol levels and reduction of the risk for chronic heart disease and stroke [35,40]. We found two studies investigating the DASH in a (sub) population with hyperuricemia but results on cardiovascular risk factors, such as blood pressure, were not specified for the patient group with hyperuricemia.

In this review, we decided to not include cohort studies and only focused on interventions because these are often more extensively described and less influenced by confounding factors. However, the intervention studies that were included differed substantially and most of them had a moderate to high risk of bias. Therefore, we did not perform a meta-analysis on the effect of the different interventions on SUA. Moreover, most studies were of short duration and lacked long-term follow-up data.

Overall, the different types of diet interventions gave a small decrease in SUA levels, but it is difficult to draw conclusions, as the quality of the studies was low with a high risk of bias. The Mediterranean-style diet seems to be the most suitable long-term diet for the targeted patient group, as it addresses both SUA as well as cardiovascular risk factors and has been more extensively studied in patients with a high cardiovascular risk (without hyperuricemia/gout). Effects of the Mediterranean diet on patients with gout or hyperuricemia, however, have not yet been studied sufficiently. A purine-low diet could be an alternative, although this diet is less comprehensive and mainly focuses on its effect on SUA. Based on the current evidence, however, a Mediterranean diet low in animal-based purines obviously adds the benefits of both diets. In the category of Mediterranean-style diets, the DASH is most probably best defined and evidence based regarding cardiovascular risk factors.

In addition, we recommend emphasizing lifestyle therapy more in guidelines for the treatment of hyperuricemia and gout. We also question whether making patients aware of the potential effects of diet and lifestyle might be easier when symptoms are still present instead of suppressed by medication. Feeling any effect of one’s own lifestyle improvement might be motivating and helpful to reduce long-term use of medication.

Since evidence is scarce and animal- and plant-based purines might have different effects on SUA, we suggest a pilot study comparing the effect of a whole food plant-based diet (plant-based purines only) compared to a purine-low diet regarding SUA, disease activity, and cardiovascular risk factors in patients with gout.

## Figures and Tables

**Figure 1 nutrients-11-02955-f001:**
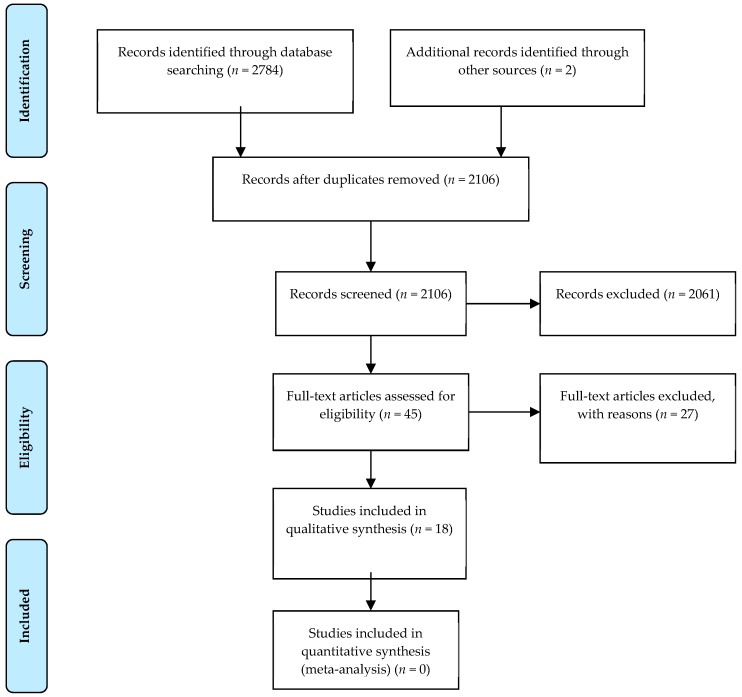
Flow diagram of the systematic literature search based on the PRISMA (Preferred Reporting Items for Systematic Reviews and Meta-Analyses) method on diet and hyperuricemia and/or gout.

**Figure 2 nutrients-11-02955-f002:**
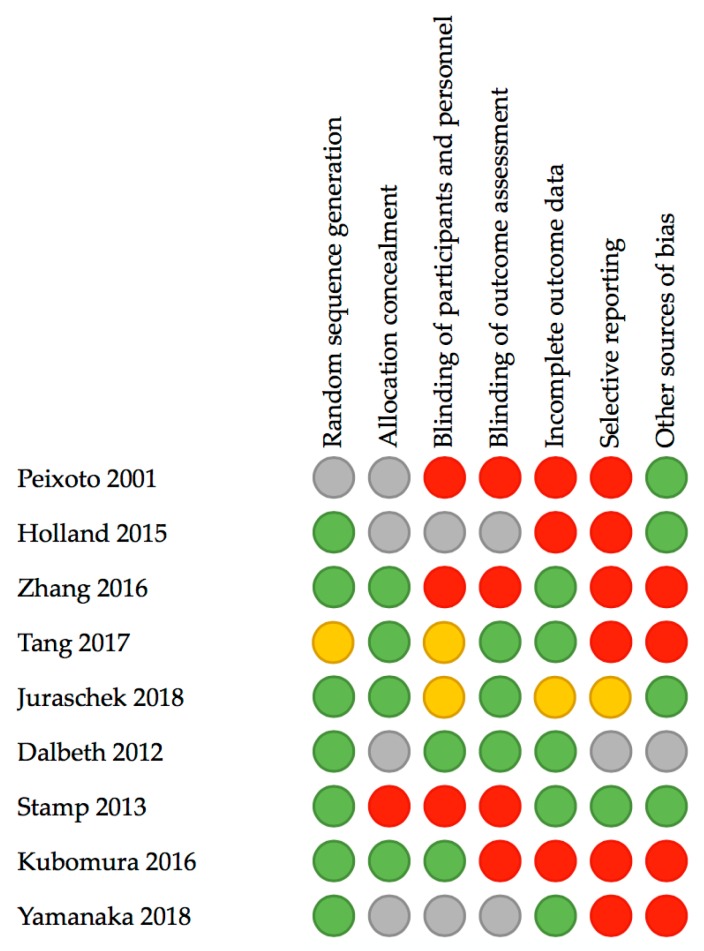
Assessment risk of bias of randomized controlled trials included in this review according to the Cochrane Collaboration’s tool. Red: high risk, yellow: moderate risk, green: low risk, grey: no information or unclear risk.

**Figure 3 nutrients-11-02955-f003:**
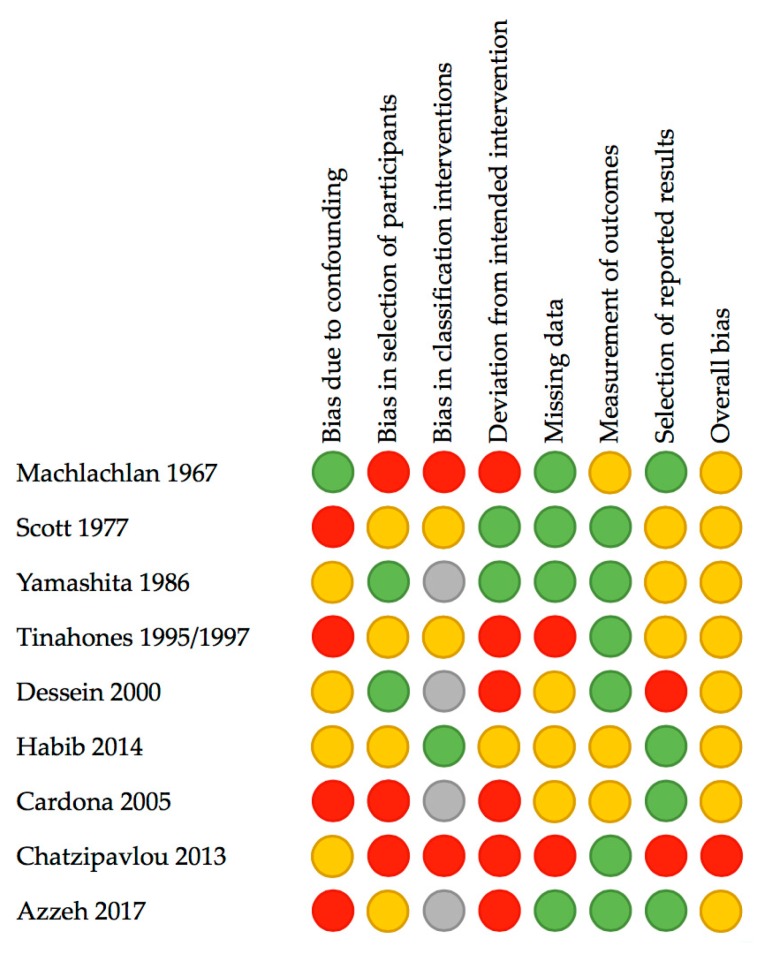
Assessment risk of bias of non-randomized studies included in this review according to the Risk of Bias in Non-randomized Studies-of Interventions (ROBINS-I) assessment tool. Red: high/critical risk, yellow: moderate risk, green: low risk, grey: no information or unclear risk.

**Table nutrients-11-02955-t001a:** (**a**)

First Author Year Journal	Study-Design Population (*n*) Groups/Control	Intervention Duration	Serum Uric Acid (SUA) Baseline mg/dL		SUA End mg/dL		Change	Remarks
			Intervention	Control	Intervention	Control	*p*-Value	
Machlachlan [14] 1967 American Journal of Medicine	Pilot study *n* = 9 (1♀) with gout Age ♀47/♂39–67 Control group: *n* = 2 normouricemic, non-gouty men	(a) 1–2 week low-purine diet (b) 1–2 days fasting (c) 1–2 days low purine diet + fasting (d) 1–2 days fasting + alcohol	7.2–13.9	4.1–5.0	Not applicable (na)	na	(a) ≈−1.4 (b) 0.5–2.1 (c) 0–0.5 (d) 0.7–2.7	No standard deviations (SDs) and *p*-values, patients were discussed individually. Results are average ranges. After fasting, SUA increased and returned to near baseline values within 24 h.
Scott [15] 1977 Advances in Experimental Medicine and Biology	Pilot study *n* = 35 (11♀) with obesity (defined as 10% above ideal weight) some with gout Control group: no	Low carbohydrate diet with 1 week low-purine (<200 mg) and alcohol-free diet before baseline measurements, Duration: until adequate weight loss	6.96		6.27		−0.69 *p* = 0.000025	Association between reduction of total cholesterol (TC) and SUA (r = 0.62, *p* = 0.004).
Yamashita [16] 1986 International Journal of Obesity	Pilot study *n* = 27 (16♀) Japanese patients BMI >26 kg/m^2^ some with gout Mean age ♀37/♂32 Control group: subjects with healthy weight	1500–800 kcal diet (stepwise reduction) Duration: 4–14 weeks	♂ 9.2 ± 1.9 ♀ 6.8 ± 1.9	♂ 5.1 ± 0.8 ♀ 4.4 ± 1.0	♂ 7.4 ± 1.6 ♀ 5.5 ± 0.9	na	*p* < 0.01	Size of control group unknown. ‘Overweight’ defined as 130% of ‘ideal weight’ (recalculated≈ BMI = 26 kg/m^2^). *p*-value: when compared with baseline values (in-group).
Tinahones [17,18] 1995/1997 Annals of the Rheumatic Diseases/Journal of Clinical Endocrinology and Metabolism	Pilot study *n* = 36 (0♀) Age 36–52 Group 1: SUA >7.0 mg/dL Group 2: SUA >7.0 mg/dL and triglycerides >2.26 mmol/L	1200 kcal diet 3 weeks	(1) 8.68 ± 1.7 (2) 8.78 ± 2.8	na	(1) 8.05 ± 1.4 (2) 7.5 ± 1.9	na	(1) −0.63 not significant (ns) (2) −1.28 ns	
Dessein [19] 2000 Annals of the Rheumatic Diseases	Pilot study *n* = 13 (0♀) Age 38–62 with ≥2 gout flares in the last 4 months, without tophi Control group: no	1600 kcal diet (The Zone Diet), 40 en% carbohydrates, 30 en% protein, 30 en% fat, no purine restriction 16 weeks	9.66 ± 1.7	na	7.97 ± 1.5	na	−1.69 *p* = 0.001	Number of self-reported monthly attacks was a primary outcome measure. Monthly attacks decreased from 2.1 ± 0.8 to 0.6 ± 0.7 (*p* = 0.002).
Habib [20] 2014 Journal of Clinical Rheumatology	Prospective cohort *n* = 43 (2♀) Age 32–82 diagnosed with gout Control group: No fasting	Ramadan: fasting including no drinking from sunrise to sunset 1 month	7.92 ± 1.69	7.6 ± 1.6	8.11 ± 1.84	7.4 ± 1.8	0.19 *p* = 0.999	Change and *p*-value: when compared with baseline values (in-group).

**Table nutrients-11-02955-t001b:** (**b**)

First Author Year Journal	Study-Design Population (*n*) Groups/Control	Intervention Duration	SUA Baseline mg/dL		SUA End mg/dL		Change	Remarks
			Intervention	Control	Intervention	Control	*p*-Value	
Peixoto [21] 2001 Arquivos Brasileiros de Cardiologia	Randomized controlled trial (RCT) *n* = 60 (31♀), *n* = 55 completed the study Age 30–75 (mean: 54.4 ± 10.6) with SUA ♂ ≥8.5 mg/dL ♀ ≥7.5 mg/dL	3 groups purine low diet purine low diet + allopurinol allopurinol only purine-low diet: restriction of products containing 100 mg purine/100 g product 12 weeks	1. 8.64 ± 1.09	2. 9.36 ± 0.89 3. 9.05 ± 1.23	1. 7.40 ± 1.27	2. 6.88 ± 1.72 3. 6.66 ± 1.73	1. −1.24, *p* < 0.001 2. −2.48, *p* < 0.001 3. −2.42, *p* < 0.001	Delta’s and *p*-values: within group changes. Between 12 and 24 weeks SUA in group 1 further decreased to 6.55 ± 2.25 (*p* < 0.001) and increased in groups 2 and 3 (*p*-value when compared to baseline <0.05).
Cardona [22] 2005 The Journal of Rheumatology	Pilot study *n* = 64 (0♀) Mean age 50 with gout according to ACR criteria, no medication for gout, hyperlipidemia, hypertension, diabetes Control group: no	Purine-low diet: no meat, poultry, fish, seafood, alcohol, beans, peas, lentils, spinach, oatmeal, asparagus 2 weeks	7.7 ± 1.7	na	7.13	na	–0.57 ± 1.7 *p* = 0.01	SD end measurement not available.
Holland [23] 2015 Internal Medicine Journal	RCT *n* = 29 (2♀, 28 completed the study) Australian patients with gout according to ACR criteria, stable on urate lowering therapy with SUA <6.05 mg/dL Age intervention group (*n* = 14, 2♀) 64 (44–80) Age control group (*n* = 16, 0♀) 61 (38–77)	Control group: general advice, compliance medication, weight loss, exercise, alcohol, target SUA Intervention group (in addition to intervention control group): additional dietary education 1 consult + follow up after 6 months	4.88 ± 1.3	4.88 ± 1.0	5.04 ± 1.2	4.54 ± 1.2	0.16 in group 0.50 between group (*p* > 0.05)	

**Table nutrients-11-02955-t001c:** (**c**)

First Author Year Journal	Study-Design Population (*n*) Groups/Control	Intervention Duration	SUA Baseline mg/dL		SUA End mg/dL		Change	Remarks
			Intervention	Control	Intervention	Control	*p*-Value	
Chatzipavlou [24] 2013 Rheumatology International	Pilot study *n* = 12, (6 completed the study, 1♀) Greek patients with asymptomatic hyperuricemia (SUA ♀ >5.7 mg/dL ♂ >7.0 mg/dL) Mean age 53 Control group: no	Cretan Mediterranean diet, high in MUFAs, legumes, cereals/bread, fruit, and vegetables; moderate in alcohol and dairy and low in meat. 8 weeks with additional measurements at 12 and 24 weeks	9.12 (8.2–13.8)	na	8 week: 6.2 (5.5–7.8) 24 week: 6.13 (na)	na	−2.92 *p* = na	SUA: averages for 6 patients with (ranges).
Zhang [25] 2016 International Journal of Food Sciences and Nutrition	RCT *n* = 187 (94♀) Chinese patients with asymptomatic hyperuricemia (SUA ♀ >6.0 mg/dL ♂ >7.0 mg/dL) Age 20–59 Control group: standard diet for hyperuricemia	Diet high in fruits and soybean products. 12 weeks	7.71 ± 1.0	7.62 ± 1.0	6.61 ± 13.3	6.62 ± 1.2	no difference between groups intervention group: −1.0, *p* < 0.001 control group: −1.0, *p* < 0.001	Standard diet for hyperuricemia: restricted energy, fat, and animal protein intake (especially red meat and seafood), less salt restriction, increased intake of soybean products, fruits and vegetables, limiting alcohol or strong tea intake, increase water intake, and moderate intake of sweet fruits and seafood.
Tang [26] 2017 Clinical Rheumatology	RCT (crossover trial) *n* = 24 (subgroup of total 103 patients with (pre) ]hypertension, 34%♀) patients SUA ≥6.0 mg/dL Age 41–61 Control group: typical American diet	DASH with 3 sodium levels 90 days	6.6 (6.3, 6.9)	6.7 (6.3, 7.1)	5.6 (4.9, 6.3)	6.6 (5.9, 7.3)	−1.02 (−2.0, −0.1) *p* = 0.03	SUA: mean with (95% CI)
Juraschek [27] 2018 Arthritis Care Res (Hoboken)	RCT (ancillary study) *n* = 117 (55% with SUA >6 mg/dL, 70%♀) African Americans with controlled hypertension Age 59 ± 9.5 Control group: DASH brochure and $30/week to purchase foods	DASH-plus: coach-directed dietary advice, assistance with purchasing DASH-related foods ($30/week), and home food delivery via a community supermarket 8 weeks	6.51 ± 1.45	6.18± 1.89	6.58 ± 1.35	6.25 ± 1.86	Change between groups −0.01 (95% CI −0.39, 0.38) *p* = 0.98	Significant trend toward greater reduction in SUA among those with a higher SUA levels at baseline (*p* = 0.008 for trend)

**Table nutrients-11-02955-t001d:** (**d**)

First Author Year Journal	Study-Design Population (*n*) Groups/Control	Intervention Duration	SUA Baseline mg/dL		SUA End mg/dL		Change mg/dL	Remarks
			Intervention	Control	Intervention	Control	*p*-Value	
Dalbeth [28] 2012 Annuals of Rheumatic Disease	Randomized double-blind controlled trial, *n* = 120 (16♀), New Zealand, patients with gout, >2 flares in last 4 months.	skim milk powder (SMP) control SMP glycomacropeptide (GMP) and G600 milk fat extract (G600) lactose powder control 3 months	1. 6.9 ± 1.5 2. 7.1 ± 1.8	3. 7.4 ± 1.8	na	na	*p*(group) = 0.15 *p*(time) = 0.27 *p*(interaction) = 0.89	Frequency of gout flares decreased significantly in all 3 groups.
Stamp [29] 2013 Arthritis & Rheumatism	Open label, parallel group, randomized controlled trial *n* = 40, (4♀) New Zealand, patients with gout, SUA >6 mg/dL, *n* = 20 without ULT (group 1 and 2) and *n* = 20 already on ULT (group 3 and 4)	500 mg Vitamin C start allopurinol allopurinol + increase dose allopurinol + 500 mg Vitamin C 8 weeks	1 + 4: 8.4 ± 1.8	2 + 3: 8.4 ± 1.5	na	na	1: −0.07 ± 0.4 2: −2.5 ± 0.4 *p* < 0.001 3: −1.5 ±0.4 4: −0.5 ± 0.4 *p* < 0.029	
Kubomura [30] 2016 Biomedical Reports	Double blind placebo controlled RCT *n* = 48, (0♀) Japanese patients with asymptomatic hyperuricemia (SUA 6.5–8.0 mg/dL) not on urate lowering therapy Age 20–64Control group: placebo	Tuna extract supplement low (238,6 mg) or high (477,1 mg) dose, spread over 3 x *p* day after each meal 4 weeks (with additional measurement at 6 weeks)	1. 7.2 ± 0.1 2. 7.2 ± 0.1	7.1 ± 0.1	1. 6.97 2. 6.86	7.03	In group change 1. −0.23 2. −0.34 *p*-value not given	In group change 0–6 weeks in high dose group −0.49 mg/dL and +0.14 mg/dL in placebo group (*p* < 0.05).
Azzeh [31] 2017 Pharma- Nutrition	Pilot study *n* = 25 (14♀) Saudi patients not on urate lowering therapy of whom 15 (6♀) with gout (group 1, age 53, BMI 31 kg/m^2^) and 15 (8♀) with hyperuricemia (group 2, age 54, BMI 33 kg/m^2^) Control group: no	500 mg vitamin C chewable tablet daily 8 weeks	1. 8.09 ± 1.09 2. 7.94 ± 0.93		1. 8.4 ± 1.15 2. 7.16 ± 1.04		1. 0.31 ± 0.14 (ns) 2. −0.78 ± 0.3 (*p* < 0.05)	
Yamanaka [32] 2018 Modern Rheumatology	Double blind placebo controlled RCT *n* = 25, (0♀) Japanese patients with asymptomatic hyperuricemia (SUA ≥7 mg/dL) not on urate lowering therapy Age 59–70 Control group: placebo (same yoghurt 2 × 100 g/day without PA-3)	PA-3Y (PA-3-containing yoghurt) 2 × 100 g/day fermentation with strains of *Lactobacillus delbrueckii* ssp. and Streptococcus thermophilus (*n* = 13) 8 weeks	8.7 ± 1.0	8.5 ± 0.9	8.7 ± 1.2	8.7 ± 1.1	No change (no difference between groups)	In group change intervention group: +0.1 ± 0.8mg/dL, in group change control group: +0.1 ± 0.7mg/dL

**Table 2 nutrients-11-02955-t002:** Overview outcomes cardiovascular risk factors of the included studies.

First Author Year	Blood Pressure in mm/Hg	Total Cholesterol in mmol/L	Triglycerides in mmol/L	High Density Lipoprotein (HDL) in mmol/L	Low Density Lipoprotein (LDL) in mmol/L	Glucose in mmol/L	Weight in Kg or Body Mass Index (BMI) in kg/m^2^
Scott [15] 1977	-	baseline: 6.3 end: 6.0 *p* = 0.03	baseline: 2.1 end: 1.5 (*p* = 0.03)	-	-	-	mean weight loss 7 kg (range weight loss 1.6–12.3 kg)
Yamashita [16] 1986	-	-	-	-	-	-	weight loss (% of baseline weight) ♂ 7.8 ± 2.4% ♀6.9 ± 2.1%
Tinahones [17,18] 1995/1997	-	1. baseline 5.2 ± 1.2; end 4.5 ± 0.9 (*p* < 0.05) 2. baseline 6.2 ± 1.1; end 5.3 ± 1.4	1. baseline 1.4 ± 0.5; end 1.1 ± 0.4 2. baseline 4.2 ± 3.8; end 1.7 ± 1.5 (*p* < 0.001)	1. baseline 1.1 ± 0.3; end 1.0 ± 0.2 2. baseline 0.9 ± 0.2; end 1.0 ± 0.3 (*p* < 0.05)	-	-	1. baseline 83.8 ± 9; end 79.8 ± 8 kg 2. baseline 85.5 ± 13; end 80.6 ± 13 kg
Dessein [19] 2000	-	Baseline 6.0 ± 1.7; end 4.7 ± 0.9 (*p* = 0.002)	Baseline 4.7 ± 4.2; end 1.9 ± 1.0 (*p* = 0.001)	Baseline 0.87 ± 0.21; end 0.91 ± 0.16 (*p* = 0.5)	Baseline 3.5 ± 1.2; end 2.7 ± 0.8 (*p* = 0.004)	-	Baseline BMI 30.5 ± 8.1; end 27.8 ± 7.9 kg/m^2^ (*p* = 0.002)
Habib [20] 2014	-	-	-	-	-	-	Ramadan group: baseline BMI 30.5 ± 5.8, end 31.0 ± 5.6 kg/m^2^; No Ramadan group: baseline BMI 29.8 ± 4.8, end 30.2 ± 5.2 kg/m^2^
Peixoto [21] 2001	1. baseline 150 ± 22.6/102 ± 17.3; end 133 ± 21.7/92 ± 12 (*p* < 0.05) 2. baseline 140 ± 21.3/95 ± 11.1; end 140 ± 20.0/95 ± 13.1 3. baseline 141 ± 23.6/92 ± 14.8; end 140 ± 19.1/93 ± 10.6	1. baseline 5.8 ± 1.3; end 6.1 ± 1.1 2. baseline 5.9 ± 1.5; end 6.4 ± 0.9 3. baseline 6.0 ± 1.5; end 6.1 ± 1.4	1. baseline 2.5 ± 0.9; end 2.6 ± 0.8 2. baseline 3.2 ± 1.3; end 3.7 ± 1.5 3. baseline 2.6 ± 1.9; end 2.4 ± 1.1	-	-	1. baseline 5.8 ± 1.0; end 5.9 ± 0.7 2. baseline 5.5 ± 1.0; end 5.5 ± 0.8. 3. baseline 5.8 ± 1.1; end 5.4 ± 0.8.	1. baseline 28.7 ± 4.2; end 28.7 ± 3.9. 2. baseline 28.4 ± 3.1; end 28.0 ± 3.0. 3. baseline 28.5 ± 4.7; end 28.2 ± 4.7
Cardona [22] 2005	-	Baseline 5.5 ± 1.0; change (2 weeks) −0.3 ± 0.8 *p* < 0.001	Baseline 2.8 ± 2.5; change (2 weeks) −0.8 ± 1.8 *p* < 0.001	-	-	Baseline 6.0 ± 1.1; change (2 weeks) -0.3 ± 0.7 *p* < 0.001	Baseline 30.2 ± 3.8, no significant change
Chatzipavlou [24] 2013	-	Average TC at baseline and 24 weeks (*n* = 6): 5.7 and 5.4	Average triglycerides at baseline and 24 weeks (*n* = 6): 1.8 and 1.3	Average HDL at baseline and 24 weeks (*n* = 6): 1.3 and 1.4	Average LDL at baseline and 24 weeks (*n* = 6): 3.5 and 3.2	-	BMI baseline 31.46 (range 28.2–37.3), at 8 weeks 29.4 (range 26–34.2) kg/m^2^, change −2.1 kg/m^2^ (*n* = 6)
Zhang [25] 2016	1. baseline 120.53 ± 11.58/78.29 ± 9.18; end 120.35 ± 13.05/77.18 ± 9.17 2. baseline 121.19 ± 14.07/77.87 ± 8.92; end 121.75 ± 15.33/77.26 ± 8.86	1. baseline 5.26 ± 1.05; end 5.08 ± 0.87 (*p* < 0.05) 2. baseline 5.20 ± 0.88; end 5.23 ± 0.88	1. baseline 1.75 ± 1.12; end 1.58 ± 0.89 (*p* < 0.05) 2. baseline 1.91 ± 1.26; end 1.85 ± 1.33	1. baseline 1.39 ± 0.36; end 1.41 ± 0.33 2. baseline 1.33 ± 0.35; end 1.41 ± 0.30 (*p* < 0.05)	1. baseline 3.05 ± 0.81; end 2.98 ± 0.73 2. baseline 3.03 ± 0.75; end 3.06 ± 0.71	1. baseline 5.45 ± 0.87; end 5.45 ± 0.55 2. baseline 5.33 ± 0.56; end 5.48 ± 0.48	1. baseline: 25.5 ± 3.4; end 25.0 ± 3.2 kg/m^2^ (*p* < 0.05) 2. baseline 25.2 ± 2.8; end 25.1 ± 2.8

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
