# Peer review of "Dietary Interventions for Gout and Effect on Cardiovascular Risk Factors: A Systematic Review"

_nutrients, 2019, doi:10.3390/nu11122955_

Round 1
Reviewer 1 Report
Excellent work.
The objective of this systematic review is to better define the most appropriate diet addressing both disease activity and traditional cardiovascular risk factors in hyperuricemic patients.
The Authors seem enough experts in regard to hyperuricemia and its related diseases, including cardiovascular risk, investigating the effect of dietary interventions (calorie restriction and fasting, purine low diets, Mediterranean style diets, DASH diet and supplements) on serum uric acid levels, gout and cardiovascular risk in two dedicate sections.
I think that the topic of this review is very important for the management of hyperuricemia in daily clinical practice.
The manuscript is very well articulated and structured in its sections; design, search strategy, keywords, eligibility criteria, material and methods and data analysis seem adequately corrected and suitable to scientific rigour; the scientific contents are supported by valid References and the narrative is comprehensive and adequately clear/understandable to common Reader; figure and tables are also clear.
I think this is a good and useful review; I have no suggestions to give to the Author nor specific comments to make.
Author Response
Thank you very much for your compliments.
Reviewer 2 Report
This was a useful review of interventional studies of diet in HU and gout. Owing to the sparse useful studies and heterogeneity this was more of a descriptive study. My comments are relatively minor:
Lines 34 to 36. The statement is incorrect. Gout is preceded by HU of equivalent level in men and women (>=7 mg/dL in both men and women). This is because, regardless of sex, monosodium urate crystals precipitate at the same concentration. Line 51. Gout is only independent of measured risk factors adjusted for in the multivariate regressions. This needs to be clarified. Lines 64-70 should be deleted. Opinionated and unreferenced. Line 75. Should be '....appropriate diet for preventing hyperuricemia and gout...' In Discussion it should be acknowledged that the interventions reviewed were severe and unlikely to be translatable to free-living humans outside of the research context. Lines 248-9. The Chatzipavlou et al study included women with urate <5.7 mg/dL Of course none of the women in this study would get gout attacks given monosodium urate crystals do not form until urate > 7 mg/dL.Author Response
Lines 34 to 36. The statement is incorrect. Gout is preceded by HU of equivalent level in men and women (>=7 mg/dL in both men and women). This is because, regardless of sex, monosodium urate crystals precipitate at the same concentration.
Thank you for this correction. We have adjusted this in the manuscript.
Line 51. Gout is only independent of measured risk factors adjusted for in the multivariate regressions. This needs to be clarified.
Thank you for this comment. We tried to clarify this in the new version.
Lines 64-70 should be deleted. Opinionated and unreferenced.
We have deleted lines 65-70. Line 64 gives a referenced answer to why plant-based purines do not seem to increase uric acid.
Line 75. Should be '....appropriate diet for preventing hyperuricemia and gout...' In Discussion it should be acknowledged that the interventions reviewed were severe and unlikely to be translatable to free-living humans outside of the research context.
This is also a good addition. We have added a sentence in the discussion.
Lines 248-9. The Chatzipavlou et al study included women with urate <5.7 mg/dL Of course none of the women in this study would get gout attacks given monosodium urate crystals do not form until urate > 7 mg/dL.
True, although only 1 patient in this study was female.
Reviewer 3 Report
The authors made a systemic review focused on the propria diets for the hyperuricemia and gout specific for the gout flare and cardiovascular disease. Four types of diets had been extensively reviewed, including calorie restriction & fasting, purine low diet, Mediterranean style diet and supplements. The aim is clear, inclusion and exclusion criteria are well defined, and the manuscript is well written.
The author screened only three supplements in this review article including tuna extract, vitamin C and PA-3Y. However, more diet supplements can be identified to decrease the level of hyperuricemia searched by PubMed and google, such as wheat gluten, Vit E, folic acid, L-argenine, curcumin, almond supplement, many polyphenols and flavonoids, etc. Suggest extensively review these supplements and categorize them by their mechanisms.
Author Response
Reviewer 3
The authors made a systemic review focused on the propria diets for the hyperuricemia and gout specific for the gout flare and cardiovascular disease. Four types of diets had been extensively reviewed, including calorie restriction & fasting, purine low diet, Mediterranean style diet and supplements. The aim is clear, inclusion and exclusion criteria are well defined, and the manuscript is well written.
The author screened only three supplements in this review article including tuna extract, vitamin C and PA-3Y. However, more diet supplements can be identified to decrease the level of hyperuricemia searched by PubMed and google, such as wheat gluten, Vit E, folic acid, L-argenine, curcumin, almond supplement, many polyphenols and flavonoids, etc. Suggest extensively review these supplements and categorize them by their mechanisms.
First, we thank you for this attentive observation. We have indeed added two articles: Dalbeth et al, 2012 (milk powder) and Stamp et al, 2013 (vitamin C). We agree that a first search gives an impression of many more studies. A closer look however shows that all the studies on mentioned supplements were not eligible for this review because of the following reasons: case studies, tested in normouricemic/non-gouty patients, tested in vitro, tested in murine models, don’t have serum uric acid or gout flares as outcomes or (f) were published after our search data.
Studies:
Added to this review:
Dalbeth N, Ames R, Gamble GD, Horne A, Wong S, Kuhn-Sherlock B, et al. Effects of skim milk powder enriched with glycomacropeptide and G600 milk fat extract on frequency of gout flares: a proof-of-concept randomised controlled trial. Ann Rheum Dis. 2012;71(6):929-34.
Stamp LK, O'Donnell JL, Frampton C, Drake JM, Zhang M, Chapman PT. Clinically insignificant effect of supplemental vitamin C on serum urate in patients with gout: a pilot randomized controlled trial. Arthritis Rheum. 2013;65(6):1636-42.
Considered but excluded from this review:
Wheat gluten:
Jenkins DJ, Kendall CW, Vidgen E, Augustin LS, van Erk M, et al. High-protein diets in hyperlipidemia: effect of wheat gluten on serum lipids, uric acid, and renal function. Am J Clin Nutr 2001;74:57–63.
Study was not performed in patients with gout or hyperuricemia.
Almonds:
Jamshed H, Gilani AU, Sultan FA, Amin F, Arslan J, Ghani S, et al. Almond supplementation reduces serum uric acid in coronary artery disease patients: a randomized controlled trial. Nutr J. 2016;15(1):77.
Study was not performed in patients with gout or hyperuricemia (although indeed the mean SUA was high, it was not an inclusion criteria) and therefor this study was not included.
Turmeric/curcumin:
Kiyani MM, Sohail MF, Shahnaz G, Rehman H, Akhtar MF, Nawaz I, et al. Evaluation of Turmeric Nanoparticles as Anti-Gout Agent: Modernization of a Traditional Drug. Medicina (Kaunas). 2019;55(1).
No intervention study and therefor not included.
Cherries:
Martin KR, Coles KM. Consumption of 100% Tart Cherry Juice Reduces Serum Urate in Overweight and Obese Adults. Curr Dev Nutr. 2019;3(5):nzz011.
Published after search data and also not performed within gout/hyperuricemic patients
Zhang Y, Neogi T, Chen C, Chaisson C, Hunter DJ, Choi HK. Cherry consumption and decreased risk of recurrent gout attacks. Arthritis Rheum. 2012;64(12):4004-11.
No intervention study and therefor not included in this review.
Vitamin E:
Seifi B, Kadkhodaee M, Zahmatkesh M. Effect of vitamin E therapy on serum uric acid in DOCA-salt-treated rats. Acta Physiol Hung. 2011;98(2):214-20.
Animal study, and therefor not included.
Omega 3:
Zhang M, Zhang Y, Terkeltaub R, Chen C, Neogi T. Effect of Dietary and Supplemental Omega-3 Polyunsaturated Fatty Acids on Risk of Recurrent Gout Flares. Arthritis Rheumatol. 2019;71(9):1580-6.
Published in 2019 (after performed search) and moreover no intervention study.
B-vitamins:
Zhang Y, Qiu H. Folate, Vitamin B6 and Vitamin B12 Intake in Relation to Hyperuricemia. J Clin Med. 2018;7(8).
Not performed in patients with gout and therefor not included.